# Sustainability of Oral Healthcare Services: A Mapping Review

**DOI:** 10.3390/healthcare13233023

**Published:** 2025-11-22

**Authors:** Diego R. Aguilar, Nathalia S. Guimarães, Alex Junio S. Cruz, Andre Luiz Brasil V. Pinto, Isabela A. Pordeus, Mauro Henrique N. G. Abreu

**Affiliations:** 1Graduate Program in Dentistry, School of Dentistry—UFMG, Belo Horizonte 31270-901, MG, Brazil; digoaguilar@ufmg.br (D.R.A.); aame0590@ufmg.br (A.J.S.C.); isabelapordeus@ufmg.br (I.A.P.); 2Department of Nutrition, School of Nursing—UFMG, Belo Horizonte 30130-100, MG, Brazil; nasernizon@ufmg.br; 3Centre for Science and Technology Studies, Leiden University, P.O. Box 905, 2300 AX Leiden, The Netherlands; a.l.brasil.varandas.pinto@cwts.leidenuniv.nl

**Keywords:** dental health services, sustainable development, environmental pollution

## Abstract

**Highlights:**

**What are the main findings?**
This mapping review identified 17 scientific reviews addressing sustainability in oral healthcare services, grouped into three main thematic axes: the 4Rs (reduce, reuse, recycle, and rethink), waste and sewage management, and sustainable barriers, practices, and policies.The findings revealed a predominant focus on waste management and resource consumption, limited integration of social and policy dimensions, and variable methodological rigor.

**What is the implication of the main findings?**
Advancing sustainability in dentistry requires improving research quality, expanding the use of life-cycle assessment, and conducting environmental auditsPromoting education, governance, and evidence-based policies to translate environmental awareness into daily clinical practice and oral health service management is relevant.

**Abstract:**

Background/Objectives: Environmental sustainability is increasingly recognized as a key component of healthcare governance, and dentistry represents a high-impact subsector due to its intensive use of materials, resources, and biosafety-driven disposables. Despite rising scientific interest, available evidence remains fragmented and methodologically heterogeneous. This study aims to systematically map the existing review-based evidence on sustainability in oral healthcare services. Methods: The protocol was prospectively registered on OSF. Narrative, scoping, and systematic reviews evaluating sustainability within oral healthcare services were eligible. Comprehensive searches were conducted in Embase, PubMed/MEDLINE, Scopus, Web of Science, LILACS, Cochrane Library, and regional databases (WPRIM, WHOLIS, BBO, BDENF, IBECS, PIE, ColecionaSUS), without language or date restrictions. Two reviewers independently screened studies via Rayyan, resolved discrepancies by consensus, and extracted descriptive and thematic data using a structured Population-Concept–Context eligibility framework. A qualitative inductive synthesis identified recurring domains, and methodological rigor was appraised with a modified 12-item AMSTAR-2 tool. Results: Of 5793 records retrieved, 17 reviews met inclusion criteria (8 narrative, 5 scoping, 4 systematic). Most publications (82.4%) were from the past five years. Three thematic axes were identified: (1) the 4Rs (rethink, reduce, recycle, reuse); (2) waste and effluent management; and (3) barriers, practices, and sustainability policies. Evidence was strongly concentrated in high-income countries, and methodological quality varied widely, with ten reviews scoring below 60% on AMSTAR-2. Conclusions: Review-based evidence on sustainable dentistry is expanding yet remains limited and operational in focus. The literature remains disproportionately centered on operational issues—primarily waste management and material consumption—while broader systemic determinants such as governance, equity, financing, and professional education receive comparatively little attention. Strengthening methodological rigor, harmonizing sustainability indicators, and advancing empirical evaluations are essential for guiding equitable and environmentally responsible oral healthcare systems.

## 1. Introduction

Environmental sustainability has become a defining challenge for global health systems, demanding structural transformation and evidence-based governance. The concept of planetary boundaries—scientifically defined ecological thresholds—underscores the urgency of preventing irreversible disruptions such as climate change, biodiversity loss, and pollution [1,2]. Health systems are central to this agenda: they both depend on and affect the environment through energy consumption, waste generation, and supply chain operations. Recognizing this interdependence, the United Nations (UN) and the World Health Organization (WHO) have established sustainability as a fundamental pillar of the 2030 Agenda for Sustainable Development [3,4].

Dentistry represents a particularly complex sector within this framework. The provision of dental care involves the intensive use of materials, water, and energy, as well as a reliance on single-use products and strict biosafety standards that—while essential—generate substantial amounts of waste [4,5]. This paradox, between safeguarding human health and mitigating ecological harm, has become a focal issue within sustainable healthcare discourse. Implementing sustainability principles in dentistry requires integrated strategies that connect clinical practice, education, and health policy.

The 4Rs framework—reduce, reuse, recycle, and rethink—provides both an ethical foundation and an operational pathway for steering dental practice toward sustainability [6]. Applying these principles entails minimizing disposable materials, optimizing workflow processes, selecting low-impact alternatives, and adopting life-cycle assessment approaches to product management. However, persistent barriers—economic, regulatory, and cultural—continue to restrict implementation, particularly in resource-constrained contexts [7,8]. Evidence indicates that environmental education, institutional leadership, and supportive policy environments are crucial for overcoming these constraints and embedding sustainability within the professional culture [9].

Despite the growing interest in environmental issues in dentistry, current scientific knowledge remains fragmented. Studies exhibit substantial variability in scope and methodological rigor, with most focusing on waste management while overlooking broader domains such as energy efficiency, sustainable procurement, and governance [10,11,12]. The lack of standardized frameworks and inconsistent reporting further hinders the translation of findings into practice. Consequently, a comprehensive synthesis of the available literature is essential to clarify existing trends, identify gaps, and guide future research and policy development.

A mapping review represents an appropriate methodological approach to achieve this goal. Unlike traditional systematic reviews, mapping reviews provide an overarching overview of the evidence landscape—identifying patterns, knowledge clusters, and research gaps rather than quantifying intervention effects [13,14]. Given the inherently interdisciplinary nature of sustainable dentistry, this approach allows a structured understanding of the field’s evolution and its interconnections with the environmental, social, and economic pillars of sustainability [15,16].

Accordingly, this study aims to systematically map the existing review-based evidence on sustainability in oral healthcare services. Specifically, it evaluates the methodological rigor and thematic scope of systematic, scoping, and narrative reviews addressing the environmental, social, and economic dimensions of sustainability. By organizing and synthesizing this body of evidence, the review seeks to (a) identify the principal thematic domains—such as waste management, resource utilization, sustainable materials, and environmental education; (b) assess the quality and consistency of the included reviews; and (c) establish key research and policy priorities to advance sustainable oral healthcare globally.

## 2. Materials and Methods

This mapping review was conducted following the PRISMA-ScR (Preferred Reporting Items for Systematic Reviews and Meta-Analyses extension for Scoping Reviews) guidelines [13] (Appendix A) and in accordance with the Joanna Briggs Institute (JBI) recommendations for evidence synthesis [17]. The protocol was prospectively registered in the Open Science Framework (OSF; registration https://osf.io/b2jke) (accessed on 10 October 2025). The guiding question of this mapping review is: How is the relationship between oral health services and environmental sustainability addressed?

To ensure methodological transparency and to align this mapping review with recognized scoping review standards, the eligibility criteria were structured using the Population–Concept–Context (PCC) framework. This approach is consistent with established methodological guidance for evidence mapping and scoping reviews, which emphasizes clarity in defining the scope and relevance of included studies [13,17].

The Population dimension included oral healthcare services, dental professionals, and populations receiving dental care. The Concept encompassed environmental, social, and economic sustainability in oral healthcare, including themes such as waste management, 4Rs principles, life-cycle assessment, resource and energy efficiency, governance, and educational strategies. The Context referred to any healthcare environment in which dental services are delivered, including clinical settings, hospitals, primary care units, and public or private systems in any geographical region.

Eligible study types included systematic, scoping, and narrative reviews. We excluded primary research studies, editorials, commentaries, conference abstracts without sufficient methodological detail, documents unrelated to oral healthcare, and reviews not explicitly addressing sustainability. A detailed summary of the PCC criteria is presented in Table 1.

Comprehensive searches were performed across Embase, PubMed/MEDLINE, Scopus, Web of Science, Cochrane Library, and LILACS, complemented by regional WHO databases (BBO, BDENF, IBECS, WPRIM, WHOLIS, PIE, and colecionaSUS). Gray literature, journal websites, and reference lists of selected articles were also screened to ensure comprehensiveness. The search strategy combined controlled descriptors (MeSH/DeCS) and free-text terms, linked with Boolean operators: (“sustainability” OR “green dentistry” OR “eco-friendly dentistry”) AND (“oral health services” OR “dentistry”) (Appendix B). No restrictions were applied regarding language or year of publication. The search was last updated in February 2025.

Two reviewers (D.R.A. and A.J.S.C.) independently screened all titles and abstracts using Rayyan QCRI. Full texts were retrieved for records meeting the inclusion criteria or deemed potentially relevant. Disagreements were resolved by consensus or consultation with a third reviewer (N.S.G.). Reasons for exclusion were recorded and summarized in a PRISMA-ScR flow diagram.

Data extraction was performed using a standardized Excel form developed for this study. The following variables were extracted: author(s), year of publication, country, journal, type of review, sustainability dimension addressed, characterization of the dental service (public/private; office/hospital; level of care—primary/secondary/tertiary). Data extraction was cross-checked by a second reviewer to ensure accuracy and completeness.

Methodological quality was assessed using a modified version of the 16-item AMSTAR-2 tool [18], adapted for narrative and scoping reviews. This modification was necessary as the included reviews were not restricted to systematic reviews with meta-analysis, requiring a focused appraisal of the systematic search and evidence synthesis process. To ensure replicability and fitness for purpose, four specific AMSTAR-2 criteria strictly related to the conduct and reporting of meta-analysis were omitted from the assessment: Criterion 11 (statistical combination methods), Criterion 12 (impact of RoB on pooled estimates), Criterion 14 (investigation of heterogeneity sources), and Criterion 15 (publication bias investigation). Thus, the critical appraisal concentrated on the 12 criteria relevant to the methodological rigor of the review process and the descriptive synthesis of evidence. Each item was scored as “Yes”, “Partial Yes”, or “No”. A quantitative score was calculated to classify the overall quality as high (>80%), moderate (60–80%), or low (<60%). The results of the appraisal were used to contextualize confidence in the synthesized evidence rather than to exclude studies.

Extracted data were analyzed thematically using an inductive-deductive approach. Three overarching categories were identified: (1) the 4Rs principles (reduce, reuse, recycle, and rethink); (2) waste and wastewater management; and (3) barriers, policies, and sustainable practices. Within each category, evidence was summarized to highlight trends, gaps, and implications for practice, education, and policy development in sustainable dentistry.

For the visual analysis of the most recurrent concepts in the included articles, VOSviewer software version 1.6.20 (Leiden, The Netherlands) was used. This tool enables the representation of the most frequent terms in the titles and abstracts of studies through co-occurrence maps. Only the visualization feature was employed to identify thematic clusters and conceptual cores. This stage complements data systematization and contributes to a graphical understanding of literature trends.

All data and methodological materials are available in the OSF registration link provided. No new experiments involving humans or animals were conducted; therefore, ethical approval was not required. Generative artificial intelligence (GenAI) tools were not used for data generation, analysis, or interpretation. Only minor grammar and formatting corrections were made using language editing tools in accordance with Healthcare editorial guidelines.

## 3. Results

The search strategy identified 5793 records across all databases. After removing 1491 duplicates, a total of 4302 titles and abstracts were screened for eligibility. Of these, 4279 records were excluded for not meeting the predefined criteria, as they were not systematic, scoping, or narrative reviews specifically addressing sustainability in oral healthcare services. A total of 23 full-text articles were assessed.

During full-text evaluation, six studies were excluded: three were primary research articles rather than reviews, two did not address sustainability in oral health services (focusing instead on broader environmental or health-system topics), and one was a conference abstract without sufficient methodological detail. Accordingly, 17 reviews met all inclusion criteria and were incorporated into the final synthesis (Figure 1) [19].

Among the included studies, eight were classified as narrative reviews, five as scoping reviews, and four as systematic reviews. Publication years ranged from 2010 to 2024, with 82.4% published in the last five years—demonstrating the recent and growing scholarly interest in the sustainability of oral healthcare services [20,21].

The three analytical axes—(1) The 4Rs; (2) Waste and Sewage Management; and (3) Barriers, Practices, and Policies—were inductively derived from the thematic analysis of the included reviews. The protocol was intentionally designed to allow flexible categorization, acknowledging the heterogeneity and conceptual diversity within the sustainability literature. During data extraction and synthesis, these three core domains consistently emerged as the most recurrent and analytically coherent groupings, enabling a structured organization of findings across environmental, operational, and policy dimensions.

This mapping review is therefore structured around three interconnected axes. The first axis, The 4Rs (Rethink, Reduce, Recycle, Reuse), encompasses strategies intended to minimize the consumption of carbon-intensive and polluting resources, optimize clinical workflows, and promote more conscious and sustainable practices within dental services. The second axis, Waste and Sewage Management, addresses the environmental impact of solid waste, liquid effluents, heavy metals, microplastics, and other pollutants released into waterways and ecosystems. The third axis, Barriers, Practices, and Policies, integrates systemic challenges including operational costs, fragmented regulation, governance gaps, and the implementation of governmental and institutional sustainability initiatives.

Because the themes overlap conceptually, individual reviews could be classified under more than one axis. Among the 17 included studies, the 4Rs were addressed in 92%, waste and sewage management in 23%, and barriers, practices, and policies in 15% of the reviews, confirming the predominance of environmentally focused approaches within the existing literature [22,23,24]. Table 2 summarizes these distributions.

Accordingly, the mapping review protocol was adapted to incorporate indirect references to the three analytical axes, acknowledging both the thematic and methodological heterogeneity present across the included studies. Because most reviews did not explicitly categorize sustainability-related elements under formal analytical domains, a broader interpretive approach was required. The qualitative analysis, therefore, relied on identifying recurring patterns of sustainable practices, organizational behaviors, and environmental strategies described within each review. Notably, 54% of the studies did not report the research setting, while dental clinics (46%) and hospital or primary care environments (31%) were the most frequently mentioned contexts. Consistent with the flexible nature of the synthesis, each review could be classified under more than one axis [25].

The visual synthesis (Figure 2) (https://tinyurl.com/2cuy5z58) (Accessed on 8 June 2025), generated through bibliometric term co-occurrence mapping, demonstrated the predominance of concepts such as “waste”, “sustainability”, and “environmental management”. This pattern highlights the continued emphasis of a large portion of the literature on environmentally centered mitigation strategies within healthcare settings. However, several thematic clusters—particularly public policy, environmental education, and technological innovation—appeared only weakly connected to everyday clinical dental practice. This disconnect likely reflects broader systemic constraints, including regulatory fragmentation, inadequate funding mechanisms, and persistent gaps in sustainability-related professional training. Rather than representing a lack of scientific relevance, this separation points to a translational gap in the field—one that future research and policy development must urgently address to enable the operationalization of sustainable dentistry in real-world settings [26,27].

These findings underscore the need for a critical and comprehensive perspective on the current body of scientific literature, revealing not only the progress achieved but also significant conceptual gaps, methodological omissions, and overlooked opportunities in the field of sustainable dentistry. The limited integration between environmental strategies, policy frameworks, and clinical practice suggests that sustainability within oral healthcare remains fragmented, inconsistently operationalized, and insufficiently addressed in existing reviews [28].

### 3.1. Axis 1—The 4Rs (Recycle, Rethink, Reuse, and Reduce)

The synthesis of the included reviews demonstrates that the application of the 4Rs within dentistry remains predominantly limited and often indirect. Across the literature, Rethink and Reduce emerged as the most consistently addressed pillars, reflecting efforts to modify clinical decision-making, decrease the use of disposable materials, and optimize workflows. In contrast, Reuse and Recycle were discussed far less frequently, largely constrained by biosafety regulations, sterilization challenges, infrastructure limitations, and operational pressures commonly observed in public health systems. Budget restrictions, inadequate technical capacity, and low institutional prioritization of environmental initiatives further hinder the integration of the 4Rs into routine clinical practice.

Among the studies that directly examined the 4Rs, one scoping review analyzing 128 documents found that only a minority explicitly addressed the complete framework, with a reduction in single-use products and their substitution by reusable alternatives emerging as the most adopted strategies [16]. In low-income settings, the lack of standardized protocols, supportive public policies, and professional training constitutes a major obstacle to implementing the 4Rs at scale [21].

Several reviews identified practical measures to mitigate the environmental footprint of dental clinics. These included the digitization of patient records, the reuse of office furniture and equipment, and, informed by Life Cycle Assessment (LCA) evidence, the replacement of disposable materials with sterilizable alternatives and the redesign of care pathways to reduce resource consumption. However, the feasibility of these strategies is heavily context-dependent. Much of the supporting evidence originates from robust and well-funded health systems—particularly the United Kingdom—whose structural conditions are not readily comparable to many other healthcare contexts, requiring careful adaptation before implementation [5,22].

Environmental responsibility was repeatedly emphasized within this axis, particularly regarding waste minimization and the reuse of materials. Persistent deficiencies in waste reduction, reuse, and recycling practices were highlighted as critical gaps that require urgent improvement in dental offices [23]. To address this shortcoming, one review proposed a waste management research model aimed at encouraging the systematic integration of the 4Rs into clinical routines [24].

Education emerged as a central theme, with several studies advocating for the incorporation of sustainability as a transversal principle in undergraduate dental curricula. Socio-environmental responsibility, according to these reviews, should be framed not as an ancillary topic but as a core ethical dimension of professional formation [25,26].

Ethical analyses within the literature introduced the concept of green dentistry, positioning it as a transformative approach to rethinking clinical practice. This perspective emphasizes the collective environmental impact of clinical decisions—particularly in public systems that deliver high volumes of care—and highlights the need to reduce consumption of energy, water, and materials, as well as to integrate sustainability principles into administrative and managerial processes [15,27,28].

Additional studies examined aspects indirectly related to the 4Rs. These included concerns about the excessive use of disposable materials (especially personal protective equipment, PPE) and the potential for safe reuse through validated decontamination protocols [29]. The environmental burden associated with plastic-based orthodontic aligners was also underscored, calling attention to microplastic release and the absence of adequate disposal policies, thereby reinforcing the need to rethink polymer use in dentistry [30].

Finally, a subset of reviews assessing sustainability in health promotion for vulnerable populations emphasized the replacement of environmentally harmful dietary habits with more sustainable ones—such as promoting access to safe drinking water in underserved regions—which simultaneously benefits oral and environmental health [31]. Other studies highlighted the need to reassess biofilm control routines in dental unit waterlines, given their implications for water consumption and microbiological quality [32].

**Table 2 healthcare-13-03023-t002:** Chart—Data extraction of included papers.

Reference (Authorship, Year)	Journal	Location	Sustainability Domain Accessed	Characterization of the Dental Service (Public/Private; Office/Hospital; Level of Care—Primary/Secondary/Tertiary)
Porteous (2010) [31]	Texas Dental Journal	Location of the primary studies was not reported	✓ Rethink ✓ Solid waste and sewage	✓ Not clearly reported for all references included. ✓ Settings reported in the main text: dental offices.
Leal (2015) [33]	Revista da ABENO	Brazil	✓ Solid waste and sewage	✓ Settings reported: Primary dental care, dental hospital, dental office.
Cataldi et al. (2017) [27]	Oral and Implantology	Primary studies from Olomouc, Bologna, Tokushima, Poland, Pakistan, Dublin,	✓ Rethink ✓ Solid waste and sewage	✓ Not clearly reported for all references included.
Duane et al. (2019) [22]	British Dental Journal	Primary studies from Ireland/United Kingdom.	✓ Reduce ✓ Reuse ✓ Recycle ✓ Rethink ✓ Policy and guidelines ✓ Solid waste and sewage	✓ Settings reported: dental offices, private dental practices, primary dental care
Khanna; Dhaimade (2019) [15]	Environment, Development and Sustainability	Primary studies from Canada, Jordan, Romania, North Bangalore, Saudi Arabia, India, UK	✓ Reduce ✓ Reuse ✓ Recycle ✓ Rethink	✓ Not clearly reported for all references included. ✓ Settings reported: dental offices, private dental practices, primary dental care, dental schools.
Duane et al. (2020) [5]	Journal Of Dental Research	Ireland/United Kingdom/USA/Canada	✓ Policies, barriers and opportunities ✓ Rethink	✓ Settings reported: private dental practices, primary dental care
Martin et al. (2021) [34]	Journal of Dentistry	Primary studies from Sweden, UK, India, USA, China, Jordan, Australia, Southeast Asia, Kosovo, Thailand, Iran, Nigeria, Malaysia, Serbia, Neal, Brazil, Pakistan, South Africa, Palestine, Canada, Turkey, Ireland, Europe, Scandinavia	✓ CO_2_ emission ✓ Reduce, Reuse, Recycle, Rethink ✓ Policy and guidelines ✓ Biomedical waste management ✓ Plastics (SUPs) ✓ Procurement ✓ Research and education ✓ Materials	✓ Not clearly reported for all included references. ✓ Settings reported: Primary dental care, dental hospital, dental office.
Martin et al. (2021) [16]	Journal of Dentistry	Primary studies from Sweden, UK, India, USA, China, Jordan, Australia, Southeast Asia, Kosovo, Thailand, Iran, Nigeria, Malaysia, Serbia, Neal, Brazil, Pakistan, South Africa, Palestine, Canada, Turkey, Ireland, Europe, Scandinavia	✓ CO_2_ emission ✓ Reduce, Reuse, Recycle, Rethink ✓ Policy and guidelines ✓ Biomedical waste management ✓ Plastics (SUPs) ✓ Procurement ✓ Research and education materials	✓ Not clearly reported for all included references. ✓ Settings reported: Primary dental care, dental hospital, dental office.
Bringmann et al. (2021) [25]	Brazilian Journal of Development	Primary studies from Brazil	✓ Reduce, ✓ Reuse, ✓ Recycle, ✓ Rethink	✓ Settings reported: private dental practices,
Ducret et al. (2022) [26]	Journal of Dentistry	Not reported	✓ Reduce ✓ Reuse ✓ Recycle ✓ Rethink ✓ Policies, barriers and opportunities	Not reported
Mahapatra et al. (2023) [21]	Res Militaris	Not reported	✓ Reuse ✓ Reduce ✓ Recycle ✓ Solid waste and sewage	Not reported
Deshmukh., et al. (2023) [28]	Acta Scientific Dental Sciences	Not reported	✓ Reduce ✓ Recycle ✓ Policies, barriers and opportunities	Not reported
Andayani, Soulissa, Danwieck (2024) [23]	Journal of Indonesian Dental Association	Primary studies from India, Portugal, Indonesia, Brazil, Pakistan	✓ Reduce ✓ Reuse ✓ Recycle ✓ Rethink ✓ Policies, barriers and opportunities	Not reported
Crystal et al. (2024) [32]	BMC Oral Health	Primary studies from India, Switzerland, Palestine, Australia, Uganda, Tanzania, Chile, UK, Canada	✓ Water ✓ Sanitation ✓ Solid waste and sewage	Not reported
Mitsika et al. (2024) [24]	Applied Sciences (MDPI)	Primary studies from Greece, Iran	✓ Solid waste and sewage	Settings reported: Primary dental care, dental hospital, dental office
Panayi et al. (2024) [30]	Journal of the World Federation of Orthodontists	Not reported	✓ Rethink ✓ Recycle ✓ Solid waste and sewage	Not reported
Walsh (2024) [29]	International Dental Journal	Not reported	✓ Reuse	Settings reported: dental clinic
Chart—Conceptualization of 4Rs				
Reduce	Minimize the amount of waste generated in the clinic. For example, reduce paper waste by double-sided printing and using digital communication.			
Reuse	Use reusable items instead of single-use or disposable products. For example, use sterilizable dental materials.			
Recycle	Convert waste materials into new materials that can be used again. Dentists must classify waste products as hazardous, non-hazardous, or objectionable and record how each is handled in the office.			
Rethink	Consider eco-friendly options when buying products for the clinic. For example, buy fewer items, choose ecologically friendly products, and purchase from companies that are environmentally aware.			
Source [16,34]				

### 3.2. Axis 2—Waste and Sewage Management

Waste management emerged as one of the most recurrent themes across the included reviews, reflecting increasing awareness of the environmental burden generated by oral healthcare services. However, the literature remains predominantly centered on solid waste, while the management of liquid effluents and sewage receives considerably less attention. This gap is particularly relevant in public healthcare settings, where high service demand, infrastructural limitations, and resource constraints require clear, feasible, and enforceable protocols to ensure adequate waste treatment. Although regulatory frameworks exist in several countries, their effective implementation within under-resourced health systems remains limited and insufficiently explored in the current evidence base [33].

The dental waste management cycle involves multiple interdependent stages—including segregation, storage, transportation, and final disposal—each requiring strict compliance with biosafety standards. Waste streams are commonly categorized into general, chemical, sharps, and infectious groups, underscoring the need for continuous professional training to guarantee adherence to regulatory requirements. In Brazil, for instance, there is a substantial gap between the norms established by the Brazilian Health Surveillance Agency (ANVISA) and their actual implementation in clinical practice, particularly regarding the development and operationalization of Healthcare Waste Management Plans [33]. Although awareness of these legal obligations exists, logistical challenges, insufficient institutional support, and limited professional training continue to hinder effective compliance, revealing a structural mismatch that compromises environmental policy effectiveness in oral healthcare [21].

Hazardous waste—including infectious, chemical, and toxic residues—comprises a significant portion of dental waste and requires specialized management in accordance with sanitary regulations [26]. Among hazardous materials, mercury from dental amalgams remains a major environmental concern. Its inappropriate disposal can lead to bioaccumulation and ecological contamination, prompting the need for stringent policies and reliable collection systems [23]. Persistent gaps in the adoption of amalgam separators exacerbate contamination risks, even in high-income countries [28]. The European Union alone disposes of an estimated 75 tons of dental amalgam annually, contributing directly to aquatic pollution [34]. In Brazil, although the transition toward mercury-free alternatives is underway, the absence of specific policies regulating the end-of-life management of amalgam waste continues to pose environmental and public health risks [33].

A further issue relates to the sharp increase in disposable plastics during the COVID-19 pandemic, driven by expanded use of single-use Personal Protective Equipment (PPE). This rise exacerbated the environmental footprint of dental services and intensified pressure on waste management systems. Additionally, orthodontic aligners represent a growing source of microplastic pollution, as their degradation and clinical use can generate polymeric residues [29]. These technologies reveal a broader and troubling pattern: the escalating use of non-biodegradable materials whose inadequate disposal amplifies environmental harm across both public and private dental settings [30].

Despite the predominance of discussions on solid waste, the management of liquid waste and effluents remains largely overlooked in the scientific literature [31]. Evidence on microbial contamination in dental unit waterlines is limited to a single review, despite its relevance for biosafety and environmental impact. The maintenance, decontamination, and operational management of dental water systems require integration into wider sustainability protocols that include water reuse, treatment, and safe disposal [15].

The persistent lack of standardized protocols and sustainability indicators in public dental services represents another major challenge. Environmental audits and Life Cycle Assessment (LCA) have been proposed as useful tools to quantify the ecological burden of dental practices, yet they remain underutilized and rarely integrated into routine service monitoring [27]. The absence of institutional accountability mechanisms further undermines environmental governance, rendering sustainable practices dependent on individual professional motivation rather than systematic implementation. Even where national regulations on healthcare waste exist, their incorporation into public dental services is inconsistent and, in many contexts, precarious [22,32].

### 3.3. Axis 3—Barriers, Practices, and Sustainable Policies

The synthesis of the included reviews reveals that barriers to sustainability in oral healthcare services extend far beyond technical or financial challenges, encompassing institutional, educational, organizational, and regulatory constraints. Across multiple studies, recurrent limitations include the absence of integrated public policies, insufficient professional training, and inadequate structural incentives to support sustainable practices—particularly within public health systems [5,15,21,22,24,25,27,28,29,30,32,33,34]. This persistent disconnect between institutional discourse and operational reality hinders the systematic implementation of environmentally responsible practices, which often remain confined to isolated initiatives carried out by highly motivated individuals rather than embedded within organizational culture.

Even in high-income countries with strong regulatory traditions, such as the United Kingdom, sustainability efforts in dentistry face ongoing resistance. Contributing factors include stringent biosafety regulations, lack of dentistry-specific environmental guidelines, and limited empirical data on the ecological impacts of routine clinical activities. The insufficient incorporation of sustainability principles into undergraduate dental curricula further perpetuates misinformation and low professional engagement [34]. These conditions are exacerbated by the methodological weaknesses identified in several reviews, including the absence of standardized approaches and unified indicators to measure sustainability in dental services—gaps that constrain both monitoring and the formulation of evidence-based public policies [25].

The literature consistently identifies the absence of standardized waste identification and segregation protocols as a major barrier. This inconsistency leads to heterogeneous data, complicating surveillance, accountability, and strategic planning. Nonetheless, the reviews highlight significant opportunities for advancement through professional education, patient engagement, and institutional innovations such as waste-to-energy initiatives and intersectoral partnerships to strengthen sustainable management systems [15]. The need for robust, enforceable guidelines is emphasized, requiring dental services to adhere to environmental regulations, adopt strict segregation and disposal protocols, and articulate a sector-wide commitment to sustainability [24].

Dental education emerges as a crucial but underdeveloped domain. Current approaches remain fragmented and sporadic, lacking curricular integration and pedagogical standardization. Several studies advocate the mandatory inclusion of sustainability content in undergraduate programs to cultivate ethical, critically oriented professionals capable of contributing to health systems grounded in equity, comprehensiveness, and socio-environmental responsibility [33]. This discussion is deeply linked to broader socio-environmental disparities, as significant segments of the population continue to lack access to clean water, sanitation, and safe waste disposal—conditions that directly shape clinical practice and oral health outcomes [32].

From an operational standpoint, the reviewed literature identifies promising institutional strategies to overcome resistance to change, including environmental audits, local sustainability committees, and the adoption of Life Cycle Assessment (LCA) tools to evaluate products and workflows [22]. However, the feasibility of such initiatives depends on stable governance structures and adequate financial support, which remain limited in many public health systems [5]. Moreover, the lack of coherent policies, fiscal incentives, and coordinated action plans continues to hinder the widespread implementation of sustainable practices, even when technical knowledge is available [28].

Regulatory issues also represent a recurring challenge. Several studies highlight the absence of specific regulations governing the safe reuse of dental materials, including PPE and polymer-based devices [29]. The standardization of reprocessing protocols has been proposed as a viable strategy to reduce waste generation, particularly given the growing reliance on non-biodegradable polymers in orthodontics, which raises significant biosafety and environmental concerns [30]. Without legal and technical support, any initiative aimed at changing clinical routines is likely to face institutional resistance.

Educational campaigns, training programs in waste management, and environmental certification initiatives in dental clinics—implemented even in resource-limited contexts—have demonstrated positive outcomes, notably reducing waste generation and enhancing professional engagement [27]. These findings reinforce the central role of socio-environmental responsibility as a foundational axis for high-quality clinical practice. Sustainable practices require not only technical solutions but also the consolidation of an institutional culture committed to environmental stewardship [21].

Additionally, governance fragmentation emerges as a key barrier. Decisions related to infrastructure, procurement, and workforce development are often made independently across administrative levels and sectors, preventing the creation of integrated and effective sustainability policies. The lack of coordination between health, environment, and education sectors has been particularly detrimental, limiting the design and implementation of coherent strategies for adopting sustainable clinical practices [32].

The critical appraisal performed using the modified 12-item AMSTAR-2 instrument demonstrated substantive variability in methodological rigor across the included reviews. Four reviews achieved scores above 80%, indicating high methodological quality and a low risk of bias, whereas three reviews scored between 60% and 80%, reflecting moderate methodological robustness. However, the majority of the studies (n = 10) received scores below 60% (Figure 3), signaling a high risk of bias and significant limitations in methodological transparency and reporting.

The predominance of low-quality reviews underscores pervasive weaknesses in the existing evidence base on sustainability in oral healthcare services. Such methodological fragility compromises the reliability, reproducibility, and transferability of the reported findings, limiting their utility for informing policy, clinical practice, and future research. This pattern demonstrates that sustainability research in dentistry remains methodologically immature, reinforcing the urgent need for more rigorous, standardized, and transparently reported review methodologies. As emphasized in previous evaluations [20], strengthening critical appraisal practices is essential for consolidating a robust and trustworthy evidence landscape capable of guiding decision-making within health systems.

## 4. Discussion

This mapping review demonstrates that sustainability in oral healthcare has been conceptualized across three interrelated axes: (1) the 4Rs framework, (2) the management of waste, and (3) the barriers, practices, and policies. Although the number of review-based publications remains limited and their methodological quality uneven, the field shows a clear and growing recognition that dentistry is not peripheral to the environmental footprint of health systems. Rather, the discipline contributes substantially to material consumption, waste generation, and energy use.

The predominance of the 4Rs in the literature reflects global pressures on health systems to incorporate environmentally responsible behaviors and resource-efficient clinical routines. Nevertheless, the thematic analysis shows a marked imbalance: while Reduce and Rethink appear frequently as conceptual anchors, Reuse and Recycle remain understudied and inconsistently addressed, particularly in low-resource settings where structural limitations hinder implementation [35]. The reuse of PPE represents a paradigmatic example of this tension. Although theoretically capable of reducing waste and lowering procurement costs, PPE reuse is constrained by non-negotiable biosafety standards established by agencies such as the CDC [36] (2024) and ANVISA [37] (2022–2023). At the same time, the environmental consequences of large-scale PPE disposal—including the release of toxic pollutants such as dioxins and furans during incineration—highlight an urgent need to reconcile biosafety with environmental protection [38].

Achieving this balance requires strengthening technological innovation, regulatory flexibility, and professional training, as emphasized in several reviews [39,40]. The growing scrutiny of disposable-based care models signals an ongoing cultural and operational transition in dentistry. Evidence increasingly points to the feasibility of reusable alternatives that, when adequately validated, offer both environmental and economic advantages. Such findings suggest that moving toward a circular dental economy is possible, but depends on institutional investment, policy alignment, and robust risk–benefit assessments [41,42].

The second axis—solid and liquid waste management—constitutes one of the most concrete and operationally relevant aspects of sustainable dentistry. Many reviews document gaps in professional understanding of waste composition and classification, leading to improper disposal practices, inflated incineration costs, and elevated environmental and occupational risks [42]. Limited adherence to segregation protocols and sparse institutional commitment to environmental monitoring further reduce opportunities for recycling and exacerbate the overall ecological impact of clinical practice [41]. Despite these challenges, emerging initiatives—such as the use of renewable energy sources within dental offices—demonstrate that environmentally efficient models of care are feasible when institutional governance, funding stability, and technical capacity are present [43].

Overall, the literature reveals a sustainability landscape that remains embryonic and unevenly integrated into dental services. Progress is occurring, but predominantly through fragmented, localized efforts rather than coordinated system-level approaches. Advancing sustainability in oral healthcare will require aligning biosafety, environmental goals, and health system governance—a triad that remains insufficiently harmonized across current evidence [41,43].

Despite the advances identified in this mapping review, the evidence also reveals significant blind spots in how sustainability is conceptualized and operationalized in dentistry. Most studies focus narrowly on technical–operational dimensions—such as waste management, material replacement, or resource consumption—while largely overlooking the broader social, political, and environmental determinants that shape the feasibility and equity of sustainable practices. Critical dimensions such as community engagement, social acceptability, distributive justice, and the relationship between sustainability and health inequities remain substantially underrepresented in the literature. Fragmented and isolated initiatives, disconnected from systemic reform, have limited capacity to institutionalize sustainability within oral health services. This gap underscores that the durability and effectiveness of sustainable interventions depend on governance structures—such as institutionalization, intersectoral participation, and political continuity—that are seldom integrated into oral health strategies [44,45].

One of the major contributions of this mapping review lies precisely in identifying this misalignment. The findings highlight the scarcity of studies integrating sustainability principles with the domains of social justice, civic participation, and critical professional training [46]. Even advanced technological proposals—such as artificial intelligence for resource optimization or smart waste-tracking systems—will remain insufficient unless embedded within coherent regulatory frameworks and adapted to local health system realities. Likewise, widely promoted sustainability measures, such as the phase-out of dental amalgam, encounter formidable barriers in countries of the Global South, where alternatives remain costly, procurement systems are fragile, and health professionals often lack adequate training and environmental awareness [47].

Moving sustainable dentistry forward, therefore, requires more than technical recommendations or incremental improvements. It necessitates explicit political commitment, targeted financial investments, strengthened professional education, and the development of specific public policies aligned with regional needs and health system capacities [46]. Advancing this agenda will also require applied research capable of generating actionable evidence that reflects the realities of public and private oral health services. Without such research, sustainability risks becoming a rhetorical aspiration rather than a transformative framework for reorganizing care models. Integrated approaches must consider not only environmental outcomes but also social equity, financial sustainability, and institutional accountability [48]. These dimensions illustrate the inherent complexity of sustainability in oral health services, demanding synergy between technological innovation, professional engagement, intersectoral governance, and political structures [47].

Within this context, the axis of barriers, practices, and sustainable policies becomes crucial for understanding the challenges and opportunities facing dentistry under conditions of accelerating climate and ecological crises. Sustainability in oral healthcare cannot be understood as an optional enhancement or a temporary trend, but rather as an ethical and intergenerational responsibility. Translating this principle into practice requires moving beyond declaratory commitments and strengthening clinical routines, management structures, and training practices aligned with environmental conservation and social justice [40]. However, the literature continues to exhibit substantive gaps. Notably, there is a marked deficiency of empirical studies that systematically evaluate the environmental impact of dental procedures, materials, and workflows—whether through population-based analyses, clinical trials, or laboratory experiments. The absence of such studies limits the precision with which the ecological footprint of dental care can be measured and hinders the formulation of evidence-driven public policies [46].

Research capable of quantifying the direct and indirect environmental effects of dental services remains rare, and there is an equally limited number of clinical trials comparing conventional care pathways with sustainable alternatives in terms of environmental impact, cost-effectiveness, and clinical outcomes. Laboratory investigations into biodegradable materials, low-impact sterilization technologies, water-efficient protocols, and renewable energy solutions also remain insufficient despite the global urgency for environmentally responsible biomedical innovation [21]. These research priorities are aligned with recommendations from international organizations such as the FDI World Dental Federation, which emphasizes the need for robust scientific evidence to sustain sustainable models of oral healthcare [49].

These gaps reinforce the need for sustained investment in applied research that reflects the realities of diverse health systems, especially in low- and middle-income countries. Sustainability must be integrated transversally into professional training, clinical guidelines, procurement systems, and institutional policies. Nevertheless, the literature demonstrates an absence of studies examining the role of continuing education, community perceptions, and social participation in shaping sustainable policies [50]. Moreover, existing sustainability policies are seldom evaluated, and few studies examine their actual outcomes, scalability, or adaptability across distinct socioeconomic contexts [48]. This lack of evaluative research undermines the capacity to refine and replicate effective strategies and perpetuates inequities in the adoption of sustainable practices [51].

Finally, this study has limitations that must be acknowledged. The emergent nature of the topic and the limited number of available reviews constrain the breadth of the findings. Although efforts were made to ensure methodological rigor, the mapping design inherently limits the depth of critical appraisal and the strength of some inferences. Additionally, gray literature and localized initiatives—often crucial sources for sustainability research—were searched, but no references were identified. It is also recognized that sustainability in dentistry is a rapidly evolving field, and new evidence may have emerged after the search period.

Promoting sustainability in dentistry requires more than eco-efficient technologies or isolated clinical adaptations; it demands a coordinated reorientation of health services grounded in political commitment, institutional responsibility, and ethical stewardship. Sustainable transformation depends on aligning three interdependent pillars: technical innovation, capable of reducing environmental impact; professional engagement, supported by environmental literacy and critical training; and robust governance frameworks, which create regulatory, financial, and organizational conditions that make sustainable practices feasible, equitable, and durable [36,37,52]. Only through the convergence of these pillars can dentistry move from declaratory intent to systemic implementation, embedding environmental responsibility into the everyday logic of care.

The findings also highlight the urgent need for empirical research capable of quantifying the environmental impacts of oral healthcare, including life-cycle assessments, environmental audits, and ecotoxicological analyses. Implementation studies and policy evaluations are especially necessary in underserved and resource-limited contexts, where sustainability strategies must be adapted to local infrastructures and sociopolitical realities. Strengthening these domains will enhance the production of actionable evidence, support evidence-informed policymaking, and promote more resilient, equitable, and environmentally responsible models of oral healthcare [53].

Ultimately, advancing sustainability in dentistry is not solely a technical challenge but a philosophical and ethical endeavor. It requires recognizing that clinical decisions extend beyond the dental chair, influencing ecosystems, communities, and future generations. Truly sustainable dentistry emerges from the interplay between scientific evidence, environmental consciousness, and a collective commitment to planetary health. Consolidating this vision will depend on long-term investments, intersectoral collaboration, and the translation of sustainability discourse into everyday professional practice [54].

## 5. Conclusions

Review-based evidence on sustainable dentistry is expanding, yet it remains limited and operational in focus. The literature remains disproportionately centered on operational issues—primarily waste management and material consumption—while broader systemic determinants such as governance, equity, financing, and professional education receive comparatively little attention. Strengthening methodological rigor, harmonizing sustainability indicators, and advancing empirical evaluations are essential for guiding equitable and environmentally responsible oral healthcare systems.

## Figures and Tables

**Figure 1 healthcare-13-03023-f001:**
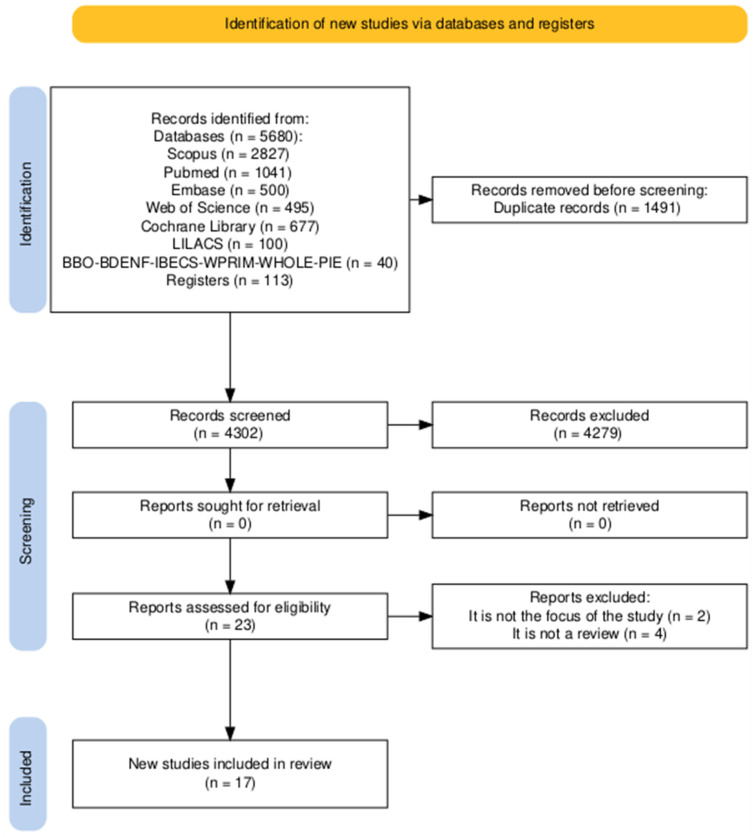
*PRISMA-ScR* flow diagram of the study selection process.

**Figure 2 healthcare-13-03023-f002:**
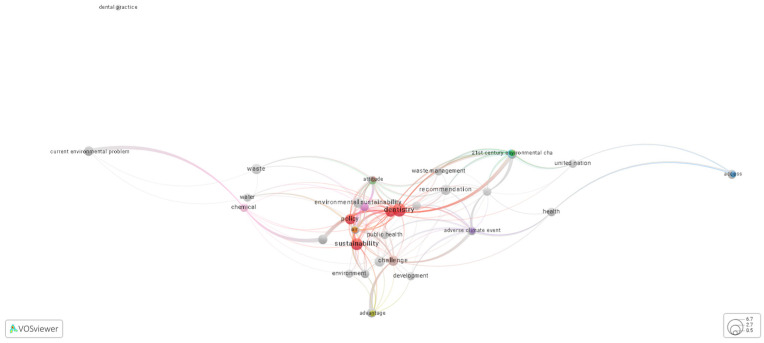
Visualization of co-occurring terms and thematic clusters in included studies (generated with VOSviewer version 1.6.20 software).

**Figure 3 healthcare-13-03023-f003:**
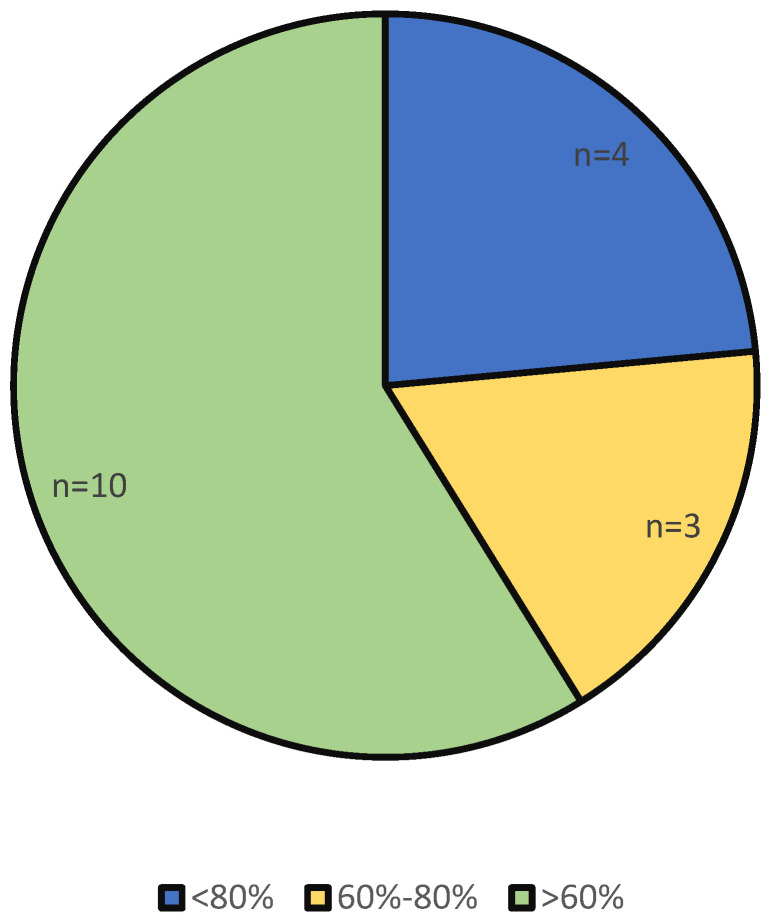
Methodological quality of the included reviews assessed with the modified AMSTAR-2 checklist (12 items), showing distribution across ≥80%, 60–80%, and <60% accuracy categories.

**Table 1 healthcare-13-03023-t001:** PCC criteria.

Component	Description	Inclusion Criteria	Exclusion Criteria
**Population**	Individuals, professionals, or systems related to oral healthcare	Oral health services; dental professionals; dental care systems; populations receiving dental care	Populations not related to dental care; general medical services without a dental focus
**Concept**	Sustainability in oral healthcare services	Environmental, social, or economic sustainability; 4Rs; waste management; LCA; governance; education; sustainable policies	Sustainability not linked to oral healthcare; environmental studies unrelated to dentistry
**Context**	Settings where oral healthcare occurs	Clinical settings, dental clinics, hospitals, primary care units, public/private systems, global contexts	Non-healthcare settings; laboratory-only environments
**Study Types**	Types of evidence included	Systematic reviews, scoping reviews, narrative reviews	Primary studies, opinion papers, commentary, conference abstracts, gray literature

## Data Availability

No new data were created or analyzed in this study.

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
