# Peer review of "Sustainability of Oral Healthcare Services: A Mapping Review"

_healthcare, 2025, doi:10.3390/healthcare13233023_

Round 1
Reviewer 1 Report
Comments and Suggestions for Authors
Dear authors,
Thank you for the opportunity to review this manuscript. The topic is timely and relevant, and the work addresses an important area of interest for the field. My detailed comments and suggestions for improvement are provided below, organized by section, in order to support further refinement of the manuscript.
Abstract
- The abstract is currently descriptive and does not clearly reflect the study aim, methods, main findings, and implications.
Recommendation: Clarify the objective, summarize the search and selection approach, report key thematic findings, and conclude with the primary implication of the review.
- Introduction
- The introduction provides extensive contextual background but does not clearly articulate the research gap or justification for conducting a mapping review.
Recommendation: Streamline this section and explicitly state (a) the gap in the existing literature, (b) why a mapping approach is appropriate, and (c) how this review contributes to the field.
- Methods
- The methodology requires greater transparency to ensure replicability.
Recommendation: Clearly specify eligibility criteria, describe the screening and data extraction process, Clarify how thematic categories were derived, may be a flowchart or a diagram.
- Discussion
- The discussion tends to repeat introductory background rather than interpret the findings in light of existing literature.
Recommendation: Strengthen interpretive depth. Clearly articulate what is novel, the implications for practice and policy, and identify remaining research gaps.
Author Response
Reviewer 1.
- Thank you for the opportunity to review this manuscript. The topic is timely and relevant, and the work addresses an important area of interest for the field. My detailed comments and suggestions for improvement are provided below, organized by section, in order to support further refinement of the manuscript.
Response: Thanks, the reviewer, for the kind assessment.
- Abstract
The abstract is currently descriptive and does not clearly reflect the study aim, methods, main findings, and implications.
Recommendation: Clarify the objective, summarize the search and selection approach, report key thematic findings, and conclude with the primary implication of the review.
Response: The abstract was fully rewritten following Healthcare guidelines. The objective is explicitly stated, methods are concisely detailed, findings quantified, and implications clearly articulated (Page 1, Abstract, paragraphs 1–3).
- Introduction
The introduction provides extensive contextual background but does not clearly articulate the research gap or justification for conducting a mapping review.
Recommendation: Streamline this section and explicitly state (a) the gap in the existing literature, (b) why a mapping approach is appropriate, and (c) how this review contributes to the field.
Response:
The introduction was restructured to clearly identify the research gap in sustainable oral healthcare, justify the mapping review approach, and articulate the manuscript’s contribution. Contextual background was streamlined (Page 2–3, Introduction, paragraphs 1–6).
- Methods
The methodology requires greater transparency to ensure replicability.
Recommendation: Clearly specify eligibility criteria, describe the screening and data extraction process, Clarify how thematic categories were derived, may be a flowchart or a diagram.
Response: Methods now include explicit PCC eligibility criteria, a detailed search strategy (databases, Boolean terms, timeframe, no language restrictions), OSF registration, dual screening using Rayyan, thematic inductive coding, AMSTAR-2 appraisal, and PRISMA-ScR flowchart. (Page 4–6, Methods, all subsections; Figure 1 (PRISMA).
- Discussion
- The discussion tends to repeat introductory background rather than interpret the findings in light of existing literature.
Recommendation: Strengthen interpretive depth. Clearly articulate what is novel, the implications for practice and policy, and identify remaining research gaps.
Response: Discussion section was rewritten to strengthen interpretation, articulating what is novel, the implications for practice and policy, and to identify remaining research gaps.
Reviewer 2 Report
Comments and Suggestions for Authors
Remove the Section "Highlights"
In such articles, Abstract requires modifications (no sub-Sections)
(135-143) Reduce that paragraph and state the Aim briefly and clearly
(196) No trade marks in Scientific articles
In general, try to reduce "Results" Section
(460-469) Reduce that paragraph
Reduce Conclusion(s) Section and state the main outcomes only
Re-write Refs according to the Journal's guidelines

Author Response
Reviewer 2
- Remove the Section "Highlights"
Response: Thank you for the opportunity to clarify this issue. The Healthcare Journal suggests the use of “Highlights”. Although it is not mandatory, we would like to maintain this section, if possible. We have modified the “Highlights” section to adhere to MDPI instructions for authors (https://www.mdpi.com/journal/healthcare/instructions). If the reviewer considers it feasible, would we maintain this section, please?
- In such articles, the Abstract requires modifications (no sub-Sections)
Response: It is possible that we would like to maintain the structured format, since the Instructions for Authors state, “Systematic reviews and original research articles should have a structured abstract.” Thanks for your understanding.
- Do not state large parts of the text using one Ref only (lines 125-134).
Response: We have modified this section.
- (135-143) Reduce that paragraph and state the Aim briefly and clearly
Response: We have reduced that paragraph and also modified the aim.
- (196) No trade marks in Scientific articles
Response: We have removed the trademarks.
- In general, try to reduce the "Results" Section
Response: We make a huge effort to reduce the Results section. In a mapping review, this section is frequently a bit larger than other types of manuscripts.
- (460-469) Reduce that paragraph
Response: We have reduced this paragraph.
- Reduce Conclusion(s) Section and state the main outcomes only
Response: Conclusion section was reduced to state the main outcome only.
- Re-write Refs according to the Journal's guidelines
Response: We double-checked all references to adhere to the Journal´s guidelines.
Reviewer 3 Report
Comments and Suggestions for Authors
Please see the document

Author Response
Reviewer 3
“Sustainability of Oral Healthcare Services: A Mapping Review”
- The manuscript addresses the concept of sustainability in oral healthcare, performing a mapping review to identify how sustainability principles have been incorporated into dental health services research. This is an emerging and relevant topic in global oral health policy and health systems research. The manuscript is well-intentioned and conceptually interesting, but methodologically limited. The paper provides a descriptive overview rather than a rigorous mapping or scoping review as defined by PRISMA-ScR or JBI methodology. Clarity on search strategy, inclusion criteria, and synthesis methods is insufficient. Consequently, the validity and reproducibility of the review are uncertain. The study tackles the intersection of oral health, public health systems, and sustainability—a relatively underexplored area. It provides a valuable conceptual foundation for integrating environmental, economic, and social sustainability into oral healthcare planning. The authors combine health policy, public health, and dental management literature. They correctly recognize that sustainability extends beyond environmental considerations to include organizational resilience, workforce stability, and access equity. The article is logically organized, with an understandable background, objectives, and thematic synthesis. The discussion section connects findings to policy implications, offering practical insights.
Response: Thanks for your revision. Each commentary will be replied below.
Major Weaknesses
- The paper does not specify the exact databases searched, search strings, date ranges, or language limits.
Response: Thank you for the opportunity to clarify these issues. In the Methods section, we have described the datasets searched (please see the first paragraph, page 4).
“Comprehensive searches were performed across Embase, PubMed/MEDLINE, Scopus, Web of Science, Cochrane Library, and LILACS, complemented by regional WHO databases (BBO, BDENF, IBECS, WPRIM, WHOLIS, PIE, and colecionaSUS). Grey literature, journal websites, and reference lists of selected articles were also screened to ensure comprehensiveness. The search strategy combined controlled descriptors (MeSH/DeCS) and free-text terms, linked with Boolean operators: (“sustainability” OR “green dentistry” OR “eco-friendly dentistry”) AND (“oral health services” OR “dentistry”) (Appendix A).
Search string is detailed in Appendix A (please see pages 19 to 23). In addition, we described on page 4, “No restrictions were applied regarding language or year of publication. The search was last updated in February 2025.”
- There is no flowchart of the selection process (e.g., PRISMA diagram) showing identification, screening, and inclusion steps. Without these details, the search process is not reproducible, undermining transparency.
Response: We have included the PRISMA diagram with all steps (see page 6). Thanks for the suggestion.
- You should provide full details of the search strategy (databases, search terms, filters, and date of last search).
Response: As we have mentioned previously, we have included all information on the search strategy. Please see pages 4 and 19 to 23.
- Include a PRISMA-ScR–style flow diagram and table of included studies.
Response: We have included the PRISMA-Scr flow diagram. The table of included studies could be found on pages 7 to 9.
- The criteria for study selection are vaguely described and not linked to a structured framework such as PICO, PCC, or SPIDER.
Response: Thanks for your suggestion. We have added PCC framework (please see page 3, second paragraph, Materials and Methods section).
- It is unclear which study types (qualitative, quantitative, policy reports) were eligible. There is no justification for excluding grey literature, which is often essential in sustainability and health policy reviews. Please define clear inclusion/exclusion criteria and provide rationale for study types considered relevant to “sustainability in oral health care.”
Response: We have modified the text to clearly address this issue (page 3, last paragraph, Materials and Methods section).
“Eligible study types included systematic, scoping, and narrative reviews. We excluded primary research studies, editorials, commentaries, conference abstracts without sufficient methodological detail, documents unrelated to oral healthcare, and reviews not explicitly addressing sustainability. A detailed summary of the PCC criteria is presented in Table 1.”
- There is no reference to following recognized scoping or mapping review frameworks (e.g., Arksey & O’Malley, Levac et al., or JBI Manual).
Response: We have included some references (please see first paragraph, Materials and Methods section).
- The review lacks protocol registration (e.g., OSF, PROSPERO).
Response: The registration of our manuscript was presented in the first paragraph, Materials and Methods section.
“The protocol was prospectively registered in the Open Science Framework (OSF; registration https://osf.io/b2jke).”
- No critical appraisal or data charting framework is reported, which limits reliability. You should align methodology with PRISMA-ScR or JBI standards, explicitly describe each stage (identification, screening, charting, synthesis), and state whether a protocol was preregistered.
Response: As we have already pointed out, we have included the PRISMA-ScR and described each stage in the Materials and Methods section. We have also included the PRISMA Scr Checklist as Appendix B.
- The results are largely narrative and descriptive, without systematic mapping of variables (e.g., publication year, country, sustainability dimension).
Response: All results are also presented in Table 2 with publication year, country, sustainability dimension, among others (please see pages 7 to 9).
- No quantitative summary or visualization (e.g., tables, thematic clusters, bubble charts) is provided. Themes are discussed but not systematically derived or validated (no coding framework described). You better add a structured data charting table summarizing each included study’s main characteristics. Use thematic or bibliometric visualization to strengthen the mapping component.
Response: Thanks, once more, for the opportunity to clarify this issue. We have performed a thematic analysis using VOS Viewer software. In the Materials and Methods section, we stated
“For the visual analysis of the most recurrent concepts in the included articles, VOSviewer software (Leiden, The Netherlands) was used. This tool enables the representation of the most frequent terms in the titles and abstracts of studies through co-occurrence maps. Only the visualization feature was employed to identify thematic clusters and conceptual cores. This stage complements data systematization and contributes to a graphical understanding of literature trends.”
In the Results section, we included Figure 2 (please see page 10), with a visualization of co-occurring terms and thematic clusters in included studies (generated with VOS viewer software).
- The discussion includes general statements about improving sustainability but lacks direct evidence from the reviewed studies. No critical comparison is made between high-income and low-/middle income countries or between clinical and policy-level sustainability.
Response: We have modified all discussion section to interpret the findings using the reviewed studies and also, included a discussion on types of countries and clinical/policy-level sustainability.
- Conclusions feel normative rather than evidence-driven. Explicitly tie conclusions to the data extracted. Distinguish between evidence-based findings and interpretive reflections.
Response: The conclusion section was rewritten to answer the main objective. Thanks once more.
Round 2
Reviewer 3 Report
Comments and Suggestions for Authors
Nothing to add